

**PeerJ Hubs**
Published on behalf of



# Infection of Atlantic tripletail *Lobotes surinamensis* (Teleostei: Lobotidae) by brain metacercariae *Cardiocephaloides medioconiger* (Digenea: Strigeidae)

Isaure de Buron[1], Kristina M. Hill-Spanik[1], Tiffany Baker[2], Gabrielle Fignar[3] and Jason Broach[3]

[1] Department of Biology, College of Charleston, Charleston, SC, United States of America
[2] Department of Pathology and Laboratory Medicine, Medical University of South Carolina, Charleston, SC, United States of America
[3] Marine Resources Research Institute, South Carolina Department of Natural Resources, Charleston, SC, United States of America

## ABSTRACT

Three juvenile Atlantic tripletail *Lobotes surinamensis* caught opportunistically in Charleston Harbor (South Carolina, USA) and maintained in captivity for over three months displayed an altered swimming behavior. While no direct causation can be demonstrated herein, fish were infected in their brain by strigeid trematode larvae (metacercariae) of *Cardiocephaloides medioconiger*, which were identified via ITS2 and 28S ribosomal RNA gene sequencing. Histology showed nonencysted metacercariae within the brain ventricle between the optic tectum and tegmentum, causing distortion of tegmental parenchyma. Aggregates of mononuclear inflammatory cells were in the ventricle adjacent to metacercariae. Metacercarial infection by *Cardiocephaloides medioconiger* has been reported from the brain and eyes of only two other fish species from the northern US Atlantic coast: the grey mullet *Mugil cephalus* and silverside *Menidia menidia,* but this identification is problematic and needs molecular verification. Atlantic tripletail is a new report as a second intermediate host for *C. medioconiger* and South Carolina is a new locality. *Cardiocephaloides* species in general have a low host specificity and infection by *C. medioconiger* could propagate to other fishes and affect neighboring natural ecosystems.

# INTRODUCTION

Digeneans are parasites with complex life cycles that involve definitive hosts, in which adults reproduce, and one to three intermediate hosts in which various larval stages (including metacercariae) develop. Most cycles involve a total of three hosts with the second intermediate host harboring metacercariae, which are trophically transmitted to their definitive hosts (*Poulin & Cribb, 2002*). In the aquatic environment, metacercariae that infect organs associated with the nervous system impact their fish host's metabolism

Corresponding author
Isaure de Buron, deburoni@cofc.edu

(*Nadler et al., 2020*) and can modify their behavior, often making them more prone to predation by their bird definitive hosts (*Lafferty & Morris, 1996*; *Seppälä, Karvonen & Valtonen, 2004*; *Fredensborg & Longoria, 2012*). These brain parasites fully integrate within food webs (*Bartoli & Boudouresque, 2007*; *Lafferty, 2008*), and all hosts involved in their life cycles can overlap, especially in confined habitats such as shallow waters, which consequently favor encounters and enhance transmission (*Combes, 2001*; *Osset et al., 2005*). Transmission can also be intensified in the wild because brain metacercariae typically display low specificity for their fish hosts (*Born-Torrijos et al., 2016*) and infect fishes with ecological similarities (*Hernandez & Fredensborg, 2015*). These brain parasites are therefore important to monitor as they can serve as bioindicators of alteration in food web dynamics and environmental disturbance (*Lafferty, 2008*; *Born-Torrijos et al., 2016*) and can negatively impact fishery and aquaculture industries (*Rosser et al., 2016*; *Palacios-Abella et al., 2018*).

The geographic range of the Atlantic tripletail *Lobotes surinamensis* (Bloch, 1790) encompasses subtropical and tropical waters of all oceans, and the fish is found in estuaries as well as offshore (*Strelcheck et al., 2004*; *Froese & Pauly, 2021*). In the US, its range extends along the Atlantic coast from New England to the Gulf of Mexico, where they are often associated with flotsam and structures such as pilings (*Strelcheck et al., 2004*). In January 2020, juvenile tripletail that had been opportunistically collected from Charleston Harbor, South Carolina (SC), USA, displayed an altered swimming behavior that was indicative of possible parasitic infection in their nervous system. While healthy tripletail exhibit a unique swimming behavior where they float on their side, mimicking floating leaves or flotsam (*Breder, 1949*), our specimens had difficulty maintaining their orientation in the water column, their tail sank below the plane of their body, and they often failed to reach the surface (Video S1). Although no causation can be certain without experimental infection, this alteration led us to suspect infection by neurotropic parasites.

Despite its broad distribution, to our knowledge only a few helminths have been reported in the Atlantic tripletail (*e.g.*, in *Baughman, 1943*; *Moravec, Walter & Yuniar, 2012*; *Dewi & Palm, 2017*) and none are metacercariae in their brain or eyes. The high quality of its flesh, the growing interest from anglers, and preliminary successful larval culture make this fish a promising candidate for mariculture in the US (*Saillant et al., 2021*). Thus, the objective of this study was to examine these fish for the presence of neutrotropic parasites that could potentially explain their altered swimming behavior. Herein, we report infection by a little-known brain parasite that has the potential to impact both the natural communities that include *L. surinamensis* and the culture endeavors of this fish species.

## MATERIALS AND METHODS

### Fish collection and maintenance and parasite collection

Three juvenile Atlantic tripletail (average 12 cm total length; range 10.7–14.2 cm) were opportunistically collected in Charleston Harbor from the Fort Johnson boat slip (32°74′27″N, 79°87′24″W) in October 2019. Fish were maintained in recirculating chlorinated/dechlorinated settled seawater in individual substrate-less glass aquaria for

three months, at which time alterations in swimming behavior were observed. Fish were euthanized *via* an overdose of tricaine methanesulfonate, MS-222 (Sigma-Aldrich, St Louis, MO) and immediately necropsied. Fish were collected, raised and euthanized by authorized staff under official permits or scientific exemptions of US state government agencies. Brains and eyes were resected and examined under a dissecting microscope. The infected brain of one individual was fixed in 10% neutral buffered formalin (NBF) and processed using standard histological techniques. Parasites were isolated from each of the two other individuals and fixed either in 95% ethanol or in sarcosyl urea for molecular identification or in 5% NBF for voucher preparation (deposited at the Museum National d'Histoire Naturelle, Paris, France under the number MNHN-HEL1889).

**Molecular identification**

DNA of one metacercaria was isolated using Sera-Mag™ Carboxylate-modified SpeedBeads (Global Life Sciences Solutions, Marlborough, MA, USA) as in *O'Donnell et al. (2016)*. DNA from a second metacercaria was extracted using a QIAGEN DNeasy blood and tissue kit (Valencia, CA, USA) following the manufacturer's protocol. Primers GA1 (5′-AGAACATCGACATCTTGAAC-3′; *Anderson & Barker, 1998*) and ITS2.2 (5′-CCTGGTTAGTTTCTTTTCCTCCGC-3′, *Bowles et al., 1993*) were used to amplify the second internal transcribed spacer (ITS2) region of the ribosomal RNA (rRNA) gene of the parasite. A portion of the large subunit (28S) rRNA gene was also amplified from the first specimen using primers LSU5 (5′-TAGGTCGACCCGCTGAAYTTAAGCA-3′; *Jensen & Bullard, 2010*) and ECD2 (5′-CTTGGTCCGTGTTTCAAGACGGG-3′; *Tkach et al., 2003*). PCR reagent concentrations and cycling followed *Hill-Spanik et al. (2021)*, as did product visualization, purification, sequencing, and sequence editing. We used the Basic Local Alignment Search Tool (BLASTN; *Altschul et al., 1990*) to compare our sequences to those in the NCBI GenBank database.

## RESULTS

All three fish examined were infected by metacercariae in the cerebellum (two in two specimens (used as voucher and for molecular identification) and one in the third fish used for histology). Microscopic examination of the serially-sectioned whole brain showed two additional free (nonencysted) metacercariae within the ventricle between the cerebellum, optic tectum, and tegmentum, both of which distorted the tegmental parenchyma. One metacercaria was more deeply invaginating the brain tissue with some sections showing ependyma-lined brain tissue present nearly circumferentially around the parasite. No definitive necrosis, gliosis, nor inflammatory infiltrate was seen within the brain tissue. However, aggregates of single cells resembling mononuclear inflammation were present adjacent to these metacercariae within the ventricle (Fig. 1). No infection occurred in the optic nerves nor within the eyes. The resulting ITS2 rRNA gene sequences (each 344 base pairs (bps)) were identical to one another and were 100% similar to a sequence of *C. medioconiger* (Dubois & Vigueras, 1949) in GenBank (accession number MN820664) collected from royal tern *Thalasseus maximus* in Mississippi, USA. The one partial 28S rRNA gene sequence (903 bp) was also 100% similar to a sequence of *C. medioconiger* in

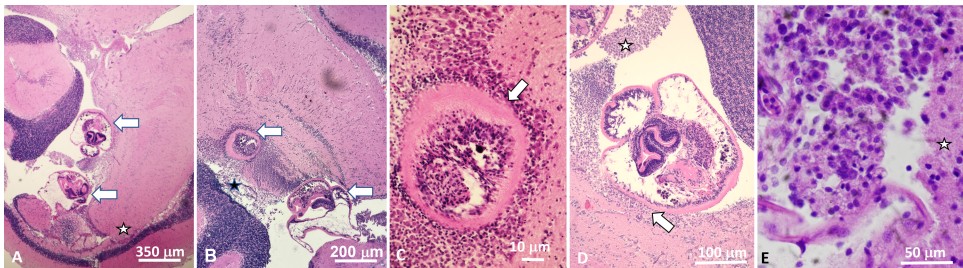

**Figure 1** Histological sections of brain of Atlantic tripletail *Lobotes surinamensis* infected with meta-cercariae of *Cardiocephaloides medioconiger* (hematoxylin and eosin). (A) Low magnification showing two metacercariae (arrows) within the ventricle between the cerebellum, optic tectum, and midbrain tegmentum (star). (B) Brain tissue is displaced and distorted around metacercariae (arrows) with no definitive necrosis, gliosis, nor inflammatory reaction within the brain parenchyma. (C) Nonencysted metacercaria (arrow) invaginating brain tissue with ependymal lining intact around the parasite. (D) Nonencysted metacercariae within the ventricle in direct contact with brain parenchyma (arrow). Aggregates of single cells closely resembling mononuclear inflammation were within the ventricle adjacent to the metacercariae (star). (E) High magnification of aggregate of mononuclear cells adjacent to metacercariae (star indicates adjacent brain parenchyma).

GenBank (MH521247), which was also collected from *T. maximus*, but in the Florida Keys, Florida, USA.

# DISCUSSION

*Cardiocephaloides* Sudarikov, 1959 is a genus of strigeid trematodes whose adults infect the intestine of marine fish-eating birds (*Dubois, 1970*). Based on the life cycle of *C. longicollis* (Rudolphi, 1819), which is the only one of seven currently accepted species of *Cardiocephaloides* whose life cycle has been fully unraveled (*Prévot & Bartoli, 1980*; *Born-Torrijos et al., 2016*), worm eggs are released in the water with the bird host feces; miracidia hatch and actively infect a gastropod first intermediate host, which releases free-living cercariae that swim, penetrate, and encyst as tetracotyle metacercariae into a fish second intermediate host, most often in the brain or eyes. The cycle is completed when birds eat infected fish. Our knowledge of the *C. medioconiger* life cycle is mostly limited to its definitive hosts: adult worms infect a variety of larid birds in Massachusetts on the northeast US coast, the Florida Keys, and the Gulf of Mexico, including the royal tern *T. maximus* (see *Dronen et al., 2007*) and several species of gulls *Larus* (see *Stunkard, 1973*; *Hernández-Mena, García-Prieto & García-Varela, 2014*; *Locke et al., 2018*; *Achatz et al., 2020*). It was also reported in the herring gull *L. argentatus* from the Republic of Korea (*Lee, Seo & Chai, 2020*).

While a few studies refer to the possibility of the nassariid eastern mudsnail *Ilyanassa obsoleta* as intermediate host of *Cardiocephaloides* in the eastern USA, they appear to be based on misidentifications of *Diplostomum nassa* (see *Martin, 1945*; *Stunkard, 1973*) (formerly *Cercaria nassa Martin, 1945*). *Hunter & Vernberg (1960)* claimed that they were successful at infecting young mullet *Mugil cephalus* with cercariae *Cercaria nassa* shed by mudsnails, and *Prévot & Bartoli (1980)* noted that these were morphologically similar
to cercariae of *C. longicollis* from *Nassa* (now *Tritia*) *corniculum* in the Mediterranean. However, while *Hunter & Vernberg (1960)* identified their specimens as *Cardiocephalus brandesi* (which later was synonymized as *C. medioconiger* (Dubois & Vigueras, 1949) Baer, 1969 - see below), these specimens were from birds fed metacercariae from naturally infected individuals of *M. cephalus* and *Menidia menidia* and not from their experimentally infected fish. *Stunkard (1973)* expressed skepticism at the validity of these authors' experimental infections and emphasized that "there is no evidence that *Cercaria nassa* is the larval stage of *C. medioconiger*" (p. 528). Lastly, *DeCoursey & Vernberg (1974)* designated cercariae also shed by mudsnails in a nearby area as *Cardiocephalus brandesii* (*Szidat, 1928*), which were later re-identified as *D. nassa* by *Sullivan, Cheng & Howland (1985)*. Therefore, the gastropod first intermediate host of *C. medioconiger* is not known, and studies of the natural history of congeneric species do not allow for accurate targeting of particular gastropods in field sampling in our area. In effect, *Donald & Spencer (2016)* reported infection by an unidentified *Cardiocephaloides* species in buccinoid whelks *Cominella* in New Zealand, while *Born-Torrijos et al. (2016)* suggested that *C. longicollis* has a narrow specificity for nassariid gastropods as first intermediate hosts.

Atlantic tripletail is a new report of a fish intermediate host for *C. medioconiger*, and this finding is particularly significant because *L. surinamensis* is being considered for extensive aquaculture in the USA (*Saillant et al., 2021*). Because the populations of intermediate hosts (and consequently their parasites' life cycles) can be amplified in aquaculture settings that may create favorable habitats (*e.g.*, *Rosser et al., 2016*), finding the gastropod host of *C. medioconiger* would allow for mitigation of the parasite in such an environment, and possibly limit infection in the wild. In effect, the lack of specificity of *Cardiocephaloides* species for their fish hosts could also be detrimental to the natural communities. For instance, *Vidal-Martínez et al. (2012)* reported *Cardiocephaloides* sp. from the Eastern Indo-Pacific infecting 9 fish species belonging to 7 families, and *C. longicollis* was reported from 31 species of 9 families (in *Born-Torrijos et al., 2016*). On the US Atlantic coast, unidentified strigeids have been reported from the brain of mummichog *Fundulus heteroclitus* along the northeastern coast (*Abbott, 1968*; *Stunkard, 1973*) and from the brain of the red grouper *Epinephelus morio* in the Gulf of Mexico (*Moravec et al., 1997*). To our knowledge the only report of a putative occurrence of metacercariae of *Cardiocephaloides* (as *Cardiocephalus brandesi*) in fish were, as mentioned above, by *Hunter & Vernberg (1960)* in the brain and eyes of grey mullet *Mugil cephalus* and silverside *Menidia menidia*. However, these authors reported their specimens as "*Cardiocephalus brandesi* (*Szidat, 1928*)", although this species was originally described by *Vigueras (1944)* and not, as they noted, by *Szidat (1928)*, who described *Cardiocephalus brandesii* (now *C. brandesii* (*Szidat, 1928*) Sudarikov, 1959), which is a valid species that also occurs in the USA (*Lumsden & Zischke, 1963*; *Dronen et al., 2007*). Because of zoological nomenclature rules, *Cardiocephalus brandesi* is an invalid species, synonymized as *C. medioconiger*. The close spelling of the two epithets may explain the inaccuracy of the descriptive authority associated with the parasite species in the report of *Hunter & Vernberg (1960)*. *Stunkard (1973)* indicated that worms collected from experimental infection of birds by these latter authors were later "submitted to Dr. Dubois and identified as *Cardiocephalus medioconiger*

[ = *Cardiocephaloides medioconiger* ]" (see p. 528), but the report by *Hunter & Vernberg (1960)* is confusing in several aspects, making the information on the natural history of *C. medioconiger* unreliable at best and demonstrates the need for further investigation of this parasite in the USA.

Pathological effects of metacercariae in a fish brain vary from severe (*e.g.*, *Dezfuli et al., 2017*) to mild (*Siegmund, Franjola & Torres, 1997*; *Grobbelaar et al., 2015*). In the tripletail infection, inflammation was limited to the ventricular space, and metacercariae appeared unaffected, as seen in other studies (*Dezfuli et al., 2017*; *Grobbelaar et al., 2015*). The mass effect exerted by the metacercariae on potentially critical tegmental regions along with the observed inflammation may at least partially explain the fish's altered swimming behavior as the proximity of the unencysted worms and inflammation to the optic tectum may have affected the processing of visual and non-visual stimuli typical of that brain region (*Northmore, 2011*). While only experimental infection would allow us to determine causation by the metacercariae of our specimens' altered swimming behavior, fish have been shown to be more predated upon when infected in their brain by various metacercariae (*Lafferty & Morris, 1996*; *Osset et al., 2005*; *Fredensborg & Longoria, 2012*). This, added to the lack of specificity of *Cardiocephaloides* for their hosts (whether definitive or intermediate), can further enhance their transmission in habitats where all hosts cohabitate such as shallow water and in areas with fisheries activity where snails and birds often congregate (*Osset et al., 2005*; *Born-Torrijos et al., 2016*). The discard of carcasses in the environment is one particular anthropogenic activity that increases transmission of this parasite (*Osset et al., 2005*; *Born-Torrijos et al., 2016*), and because it is a common practice in Charleston Harbor, it further emphasizes the need for angler education efforts to limit this and other parasite infections in our area (*de Buron et al., 2017*).

In conclusion, the little-known pathogenic parasite *C. medioconiger* was found to infect Atlantic tripletail *L. surinamensis,* a fish with ecological and economic relevance. The low specificity of *Cardiocephaloides* for its hosts, the fact that the infection could induce an impediment in the ability of the fish to escape predators, and discard of carcasses are factors that could lead to the amplification of parasite transmission in the wild. It is important to fully unravel and describe this parasite life cycle, and a survey of wild Atlantic tripletail would provide a better understanding of its geographic distribution.

## ACKNOWLEDGEMENTS

Thanks to Matt Walker at SC-DNR for assistance with DNA isolations and Mary Ann Taylor for bringing these sick fish to our attention. Fish were euthanized by authorized SC Department of Natural Resources staff under scientific exemptions of US state government agencies.

### Funding

The authors received no funding for this work.

## Competing Interests

The authors declare there are no competing interests.

## Author Contributions

- Isaure de Buron conceived and designed the experiments, performed the experiments, analyzed the data, prepared figures and/or tables, authored or reviewed drafts of the article, and approved the final draft.
- Kristina M. Hill-Spanik conceived and designed the experiments, performed the experiments, analyzed the data, prepared figures and/or tables, authored or reviewed drafts of the article, and approved the final draft.
- Tiffany Baker conceived and designed the experiments, performed the experiments, analyzed the data, prepared figures and/or tables, authored or reviewed drafts of the article, and approved the final draft.
- Gabrielle Fignar performed the experiments, authored or reviewed drafts of the article, collected fish in wild, maintained fish in captivity, made IdB aware of alteration of fish behavior, euthanized fish (authorized DNR personnel while authors from CofC had no IACUC for this fish), participated in dissection, and approved the final draft.
- Jason Broach conceived and designed the experiments, authored or reviewed drafts of the article, collected fish in wild, maintained fish in captivity & provided info for ms, euthanized (same reason as above)- his study could not have happened without the two DNR co-authors - and approved the final draft.

## Animal Ethics

The following information was supplied relating to ethical approvals (i.e., approving body and any reference numbers):

Fish were collected, raised and euthanized by authorized Department of Natural Resources (DNR) staff under official permits or scientific exemptions of U.S. state government agencies.

## Data Availability

The sequences are available at GenBank: ON815613–ON815614 (ITS2) and OP761874 (28S).

The voucher specimen was deposited at the Museum National d'Histoire Naturelle, Paris, France: MNHN-HEL1889.

## Supplemental Information

Supplemental information for this article can be found online at http://dx.doi.org/10.7717/peerj.15365#supplemental-information.

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
