# Peer review of "Infection of Atlantic tripletail Lobotes surinamensis (Teleostei: Lobotidae) by brain metacercariae Cardiocephaloides medioconiger (Digenea: Strigeidae)"

_PeerJ, doi:10.7717/peerj.15365_

## Round 0.1 · original submission · Major Revisions

This is a very interesting paper, however the fish number is very low. Please focus your discussion on the actual findings of the study and remove speculations. Please carefully address all comments from both reviewers and make all necessary changes.

·

Basic reporting

The paper is within scope and of acceptable English standard, with relevant figures. The the major literature is referenced, though the authors seem to have a poor grasp of the discipline of taxonomy. The paper describes the molecular identification of the metacercaria found and the histology of the lesion, making a valuable contribution to these topics.

However, the arguments for the wider relevance of the work to aquaculture are flimsy and could be extensively edited. Even the finding of 3/3 infected animals is no indication of wider prevalence. Infected fish are known to school together and parasite prevalence can vary seasonally by orders of magnitude. By all means argue that a wider survey be carried out, but to try and link the argument to aquaculture (at present confined to recirculation systems) is an exaggerated claim. Removing much of that argument would reduce the references cited.

Lines 37-38 “ Digeneans are parasites with complex life cycles that involve definitive hosts and one or two intermediate hosts, in which larvae (including metacercariae) develop” is incorrect. Digenea have complex life cycles that may include up to four hosts, three distinct generations and different infection strategies (Cribb et al. 2003).

Line 38-39 “ Metacercariae encyst in the second intermediate host and are always trophically transmitted to their definitive hosts” Best delete this statement. They can also be transmitted via a further intermediate host and may or may not end up in a final definitive host.

Line 44 “fully integrate food webs” Should read “fully integrate within food webs”

Line 50-51 Desfuli et al 2007 doesn’t mention aquaculture? Palacios-Abella et al., 2018 do – but they are dealing with Aporocotylids (blood flukes) whose definitive host is the fish. In their case the cercariae from the polychaete worms are directly infecting the definitive fish host that, in turn infects the polychaetes under the sea cages. If Atlantic triplefins are cultured in recirculating tanks in artificial seawater (Saillant et al. 2021), that would not be an issue.

Experimental design

There are no problems with the methodology.

Validity of the findings

Line 156 - 167 – I do not see where there is confusion. Hunter and Vernberg (1960) experimented with Cardiocephalus brandesi, Szidat 1928 collected from Terns and skimmers (Rhynchops nigra nigra.) and concluded that the cercaria, which infected mullet were morphologically identical to Cercaria nassa Martin, 1945. C brandesi was transferred to Cardiocephaloides by Sudarikov, 1959 to become Cardiocephaloides brandesii (Szidat, 1928) Sudarikov, 1959. Presumably Hunter and Vernberg had not caught up with the name change, given that in 1960 information was transmitted by mail and collated books of searchable authors and titles were distributed annually. Also, C. brandesii is a parasite of terns (Dronen et al. 2007). See https://marinespecies.org/aphia.php?p=taxdetails&id=827412 for summary.
Cardiocephalus brandesii is a synonym of Cardiocephaloides brandesii (spelt with two “ii”’s as the Latin gender of the species name and genus name must be the same ) - see https://marinespecies.org/aphia.php?p=taxdetails&id=827502.

Lines 175-187 The parasite has a wide distribution in the wild, given the distribution of the larid definitive hosts. It is presumably limited only by its snail intermediate host(s). I’m not sure why the lack of specificity in Cardiocephaloides should have any detrimental effect on natural communities that, presumably, have already been exposed for decades. For aquaculture to have any effect, you would need to show that the methodology was amplifying the number of infected gulls but, with recirculating tank culture that is unlikely to happen.
Line 184-187 There is no taxonomic confusion.

Lines 196-198 you write: “ Because brain parasites can impact the behavior of their fish hosts, they can become more prone to predation (Lafferty & Morris, 1996; Fredensborg & 198 Longoria, 2012; Osset et al., 2005).

- see lines 40-43 These read: “ metacercariae that infect organs associated with the nervous system impact their fish host’s metabolism (Nadler et al., 2020) and can modify their behavior, often making them more prone to predation by their bird definitive hosts (Lafferty & Morris, 1996; Seppälä, Karvonen, & Valtonen, 2004; Fredensborg & Longoria, 2012).”
This is repetition.

Lines 198-203 is repeating a tenuous argument made earlier.

Lines 203-204 could well be introduced earlier, here it is an afterthought to bolster a weak argument.

Lines 205-206 This is the first mention of “anglers” see comments above about a weak argument! Why should angling, an activity that has been going on for millennia, be affected by your life-cycle discovery?

Lines 210-211 is that argument made again. You have to show that there is a credible effect and that would be a different paper. This one should simply be recording what you actually found.

Line 213-214 is perhaps the most important ecological conclusion. However, be aware that parasite prevalence can vary seasonally by orders of magnitude.

I have not checked the references, but note that “parasitology” is miss-spelt in line 235.

References cited:
Cribb, Thomas H, Bray, R.A., Olson, P.D., Timothy, D. Littlewood, J. 2003. Life Cycle Evolution in the Digenea: a New Perspective from Phylogeny, Advances in Parasitology, Academic Press, Volume 54, Pages 197-254,

Dronen NO, Blend CK, Gardner SL, Jimenez FA. Stictodora cablei n. sp.(Digenea: Heterophyidae) from the royal tern, Sterna maxima (Laridae: Sterninae) from Puerto Rico and the Brazos County area of the Texas Gulf coast, USA, with a list of other endohelminths recovered in Texas. Zootaxa. 2007 Mar 26;1432(1):35-56.

Additional comments

Concentrate on the science. You have found a new record for the metacercaria, you have competently identified it by molecular methods and described the lesion. You have also linked your molecular and histology results to the wider literature. That's all you need to do.

Reviewer 2 ·

Basic reporting

Information on Cardiocephaloides species is really little, and therefore, this research is of great value as it adds new data on the occurrence of a Cardiocephaloides species on a new second intermediate host. The authors have done a good work with the limited material they had, only 3 fish, with only one or 2 metacercariae each. They managed to identify the brain area where the metacercariae were located by histological sections, and do molecular work to identify them. All methods employed are correct. However, it is important to highlight the tiny sample number and the need to continue further work to confirm the conclusions with a higher sample size. Similarly, the statements on the behavioural changes need to be carefully reworded, as no comparisons with uninfected individuals nor behaviour experiments have been done. Information on this new record and potential effects is worthy to be published, as therefore I recommend a major revision of the MS. I have listed some suggestions that should be considered:
- L. 26. “…low host specificity.”. Please, add “at the second intermediate host” or something similar to make sure the low host specificity is at the fish level (as stated later in the main text).
- L. 77. Please add more information on the total length, at least minimum and maximum is SD seems unnecessary for three samples. However, consider increasing sample size.
- L. 77. It would improve the strength of the work if the authors would increase the sample size. At least to dissect the brains (in fresh), and provide more information on the prevalence and intensity of infection.
- L. 84. What do the authors mean by “resected”? Please, explain.
- L. 91 and 93. Please correct as “metacercaria” (in singular).
- Figure 1: “Non-encysted metacercaria (arrow) invaginating brain tissue with ependymal lining intact around the parasite”. Do the authors mean that the metacercaria is “sucking” on the brain tissue? I do not observe this in the picture. Please, provide another picture where this is visible, or otherwise please remove this information, also in the text (L. 120).
- L. 123, and Fig. 1 D (also in abstract L. 21). Is it possible to provide pictures from uninfected individuals (or areas of the brain that were not affected, as the other ventricle from the same fish) to show that there are no aggregates of single cells/mononuclear inflammation?
- L. 80-81. The swimming behaviour observed in the video seems quite odd, however, I am not familiar with the normal behaviour of Atlantic tripletail. Could the authors describe the changes that they observed after the three months in the aquaria? How do the authors explain that these changes occurred only after three months?
- L. 130. Although not strictly necessary due to the 100% similarity, I personally prefer that newly produced sequences of new records are submitted to GenBank. Please, do so unless there is some justified reason.
- L. 189-195. Usually, changes in behaviour occur with higher numbers of encysted metacercariae. Could it be possible that the changes are due to a different reason? Did the authors have uninfected fish to compare this? Furthermore, even in changes in behaviour have been observed in fish with brain-encysting parasites, to conclude that these are caused by the parasite, behavioural experiments (with infected and uninfected fish) need to be performed. Personally, I can believe that Cardiocephaloides might provoke these changes, but these questions (i.e. low number of infecting metacercariae, lack of comparison to uninfected fish, lack of behavioural experiments) need to be clearly stated in the discussion. Also please modify accordingly L. 17 in the abstract (and L. 61 in Introduction) stating the changes in behaviour in comparison with uninfected individuals. Otherwise, please remove this from the abstract.
- L. 190. What do the authors mean by impingement of the metacercariae?
- L. 191-192 and 209-210. Please be careful with such statements. It might be true, but there are not enough data to state this (see comment earlier), please reword.

Experimental design

There is no experiment per se, but the employed methods are correct. Rigorous investigation has been performed, although sample size should be increased to improve the strength of the work.

Validity of the findings

As stated before, the number of samples is very low and shoudl be increased to improve the strength of the study. Furthermore, to state that parasite-induced changes are caused in a host, it is necessary to do behaviour experiments and compare infected and uninfected individuals. This is lacking, and therefore, some parts of the text (in the abstract, introduction and discussion) should be reworded to show that additional data are needed to demonstrate changes in fish behaviour. Some suggestions have been provided earlier.

---

## Round 0.2 · Minor Revisions

Congratulations on this interesting paper. Please note that both reviewers still have some comments, hope those are helpful.

·

Basic reporting

The authors have addressed the comments of the referees and with one exception I am happy with the corrected manuscript.
The issue outstanding is their assertion that there is taxonomic confusion, and this confusion was explained by them in their reply to my failure to see any confusiion, Their reply (in part) was "Below we ‘dissect’ this comment and address each point to explain and attempt to convince the editor that there is indeed taxonomic confusion, which is necessary to point out in the manuscript. Significantly, see https://www.marinespecies.org/aphia.php?p=taxlist
- Cardiocephalus brandesi (one ‘i’) was described by Vigueras (1944),
- and Cardiocephalus brandesii (two ‘ii’) was described by Szidat (1928).
The title page of Hunter & Vernberg (1960) (below) shows the root of the issue upfront as they mention Cardiocephalus brandesi (one ‘i’) with Szidat, 1928 as the descriptive authority, i.e., there is an incorrect association of parasite species and descriptive authority.

I would point the authors to the International Code of Zoological Nomenclature 4th edition, Rule 33.4 Available at https://code.iczn.org/

33.4. Use of -i for -ii and vice versa, and other alternative spellings, in subsequent spellings of species-group names
The use of the genitive ending -i in a subsequent spelling of a species-group name that is a genitive based upon a personal name in which the correct original spelling ends with -ii, or vice versa, is deemed to be an incorrect subsequent spelling, even if the change in spelling is deliberate; the same rule applies to the endings -ae and -iae, -orum and -iorum, and -arum and -iarum.
Example. The subsequent use by Waterhouse of the spelling bennettii for the name established as Macropus bennetti Waterhouse, 1837 does not make the subsequent spelling an available name even if the act was intentional.

Its a small point, based on long ago when scientists were expected to be Latin and Greek scholars, that the Latin/Greek gender of the species name must be the same as the Latin/Greek gender of the Genus name. The species ending often changes when the Genus is changed or when there is a dispute about the "correct" gender. Its always been a curse for those of us naming species in the 20th and now 21st century. The full name of a species includes the authority which is always the earliest authors name separated from the year by a comma as in Cardiocephalus brandesii Szidat, 1928. If the Genus is changed, then that is reflected by putting the original authors name in brackets, followed by the name and date of the person who made the change - as in Cardiocephaloides brandesii (Szidat, 1928) Sudarikov, 1959. The animal described by Vigueras (1944), was a misidentification of C. brandesii, or a nomen nudum (because the name was already taken), either way it has been corrected by giving it a new name by Duboise & Vigueras 1949 and genus by Baer (1969).

Experimental design

Fine

Validity of the findings

Fine

Additional comments

None

Reviewer 2 ·

Basic reporting

I would like to thank the authors for their effort to eliminate speculative information on the fish behavioural data. Having said that, I believe that L. 77-79 is still confusing, and should be reworded as “Fish were maintained in recirculating chlorinated/dechlorinated settled seawater in individual substrate-less glass aquaria for three months, during which alterations of their swimming behavior were observed”. The rest of the manuscript reads well and would be ready for publication after this change.

Experimental design

This was already evaluated and revised.

Validity of the findings

This was already evaluated and revised.

---

## Round 0.3 · accepted · Accept

Thank you for addressing all the comments, I have assessed the revision and I am happy with the current version. The manuscript is now ready for publication.